# Coral Reef Resilience in Taiwan: Lessons from Long-Term Ecological Research on the Coral Reefs of Kenting National Park (Taiwan)

**Shashank Keshavmurthy** [1,†], **Chao-Yang Kuo** [1,†], **Ya-Yi Huang** [1], **Rodrigo Carballo-Bolaños** [1,2,3], **Pei-Jei Meng** [4,5,*], **Jih-Terng Wang** [6,*] and **Chaolun Allen Chen** [1,2,3,7,8,*]

[1] Biodiversity Research Center, Academia Sinica, Taipei 11529, Taiwan; coralresearchtaiwan@gmail.com (S.K.); cykuo.tw@gmail.com (C.Y.K.); lecy.yhuang@gmail.com (Y.Y.H.); rodragoncar@gmail.com (R.C.)
[2] Biodiversity Program, Taiwan International Graduate Program, Academia Sinica, Taipei 11529, Taiwan
[3] Department of Life Science, National Taiwan Normal University, Taipei 10610, Taiwan
[4] National Museum of Marine Biology/Aquarium, Pintung 94450, Taiwan;
[5] Insititute of Marine Biodiversity and Evolution, National Dong-Hua University, Hualien 97447, Taiwan
[6] Department of Biotechnology, Tajen University, Pingtung 90741, Taiwan
[7] Institute of Oceanography, National Taiwan University, Taipei 10617, Taiwan
[8] Department of Life Science, Tunghai University, Taichung 40704, Taiwan
†: Equal contribution
**\*** Correspondence: cac@gate.sinica.edu.tw (C.A.C.); jtw@gmail.com (J.T.W.); pjmeng@nmmba.gov.tw (P.J.M.)

**Abstract:** Coral reefs in the Anthropocene are being subjected to unprecedented levels of stressors, including local disturbances—such as overfishing, habitat destruction, and pollution—and large-scale destruction related to the global impacts of climate change—such as typhoons and coral bleaching. Thus, the future of corals and coral reefs in any given community and coral-Symbiodiniaceae associations over time will depend on their level of resilience, from individual corals to entire ecosystems. Herein we review the environmental settings and long-term ecological research on coral reefs, based on both coral resilience and space, in Kenting National Park (KNP), Hengchun Peninsula, southern Taiwan, wherein fringing reefs have developed along the coast of both capes and a semi-closed bay, known as Nanwan, within the peninsula. These reefs are influenced by a branch of Kuroshio Current, the monsoon-induced South China Sea Surface Current, and a tide-induced upwelling that not only shapes coral communities, but also reduces the seawater temperature and creates fluctuating thermal environments which over time have favoured thermal-resistant corals, particularly those corals close to the thermal effluent of a nuclear power plant in the west Nanwan. Although living coral cover (LCC) has fluctuated through time in concordance with major typhoons and coral bleaching between 1986 and 2019, spatial heterogeneity in LCC recovery has been detected, suggesting that coral reef resilience is variable among subregions in KNP. In addition, corals exposed to progressively warmer and fluctuating thermal environments show not only a dominance of associated, thermally-tolerant *Durusdinium* spp. but also the ability to shuffle their symbiont communities in response to seasonal variations in seawater temperature without bleaching. We demonstrate that coral reefs in a small geographical range with unique environmental settings and ecological characteristics, such as the KNP reef, may be resilient to bleaching and deserve novel conservation efforts. Thus, this review calls for conservation efforts that use resilience-based management programs to reduce local stresses and meet the challenge of climate change.

**Keywords:** Taiwan; coral reef; marine national park; nuclear power plant; community dynamics; Symbiodiniaceae; long-term ecological data

## 1. Introduction

### 1.1. Coral Reef Ecosystems and the Impacts of Environmental Change

Coral reefs are one of the world's most diverse and productive marine ecosystems, providing essential goods and services for millions of people. A remarkably high diversity of species and interactions makes this ecosystem particularly sophisticated and sensitive to human-induced perturbations [1]. Overfishing, pollution and habitat destruction are among the most common local stressors altering coral reef ecosystem dynamics (reviewed in [1–4]). In addition, the increases in seawater temperature and ocean acidification are considered to be the two major global stressors responsible for the worldwide degradation of coral reef health (reviewed in [3]). Seawater temperatures 0.5–1.0 °C above summer average persisting for days to weeks could result in disruption of mutualistic relationship between coral hosts and their symbiotic algae (also known as zooxanthellae, family Symbiodiniaceae), resulting in "coral bleaching"; if this persists for months, it can result in mass coral mortality [5]. Several major severe bleaching episodes have been recorded since 1979. Among them, a 1998 event had the most devastating known effect, impacting over 75% of reefs worldwide, and wiping out nearly 16% of them [6]. In addition, between 2014 and 2017, "back-to-back" thermal anomalies occurred on the northeastern coast of Australia, causing massive coral bleaching in the north and mid sections of the Great Barrier Reef; this is arguably the worst-ever bleaching in the history of the GBR [7,8] and other parts of Australia [9,10]. Similar global-scale coral bleaching events (GCBE) that result in high coral mortality, the rapid decline of reef structures, and unprecedented environmental impacts have also been reported in the Indian [11,12], Pacific [13–15], and Atlantic Oceans [16,17]. Scientists have therefore concluded that the 2014–2017 GCBE represents the first multi-year, global-scale coral bleaching event to cause bleaching and mortality two or more times over the 3-year event [18].

Ocean acidification—the ongoing decrease in the pH of the ocean caused by the uptake of anthropogenic carbon dioxide ($CO_2$)—is, on the other hand, decreasing the calcification, reducing coral growth, and limiting reef development. There appears to be no chance of maintaining atmospheric $CO_2$ concentrations under 450 ppm [19] and limiting global temperature increase to less than 2 °C [20] by the end of the century; however, corals and coral reef communities at low and high latitudes might demonstrate exceptional acclimatization and adaptation capacities to survive the future environmental changes [21], although it has been suggested that most species will fail to develop mechanisms to survive future conditions, and a worldwide decline in coral reefs now seems inevitable [3]. Recent reports have shown that recurrent and prolonged thermal anomalies above the threshold limit corals' resistance to stress [22] and leads to increased bleaching, mass destruction of coral reefs, and the loss of many coral species [7,8]. The latest IPCC reports show an imminent threat to tropical coral reefs as soon as 2030 if carbon emissions continue to increase and subsequently increase average seawater temperatures. It was concluded at 1.5 °C global heating, the world will lose 70%–90% of its coral reefs, but at 2 °C, virtually all of the world's coral reefs will be lost [19].

In addition to driving more intensive mass coral bleaching, warming oceans will also enhance the destructive potential of tropical storms, including typhoons in the West Pacific, hurricanes in the Caribbean, and cyclones in the South Pacific and Indian Oceans, by either increasing their frequencies [23] or intensities [24], although other studies suggest that the global frequency of cyclones might remain stable or even decrease by up to 40% under greenhouse conditions by the end of this century [25–27]. Despite these projections, future impacts remain debatable. The future climate might, in turn, be influenced by some of the most severe ecological impacts on coral reefs through direct physical disturbances, turbidity, sedimentation, or salinity changes that would result from the destruction of reef structures [28]. As a consequence, the diversity and biomass of fish and other fauna that require corals for shelter or food will be dramatically reduced [29].

*1.2. Coral Reef Resilience under the Impacts of Environmental Change*

Resilience, a theory introduced to describe how ecosystems respond to disturbances [20], has recently been applied to coral reef ecosystems to examine how organisms respond to and interact as the result of local and global stresses. Many studies [30–35] including [36] have developed different definitions of resilience when applied to coral reef ecosystems including that the resilience is influenced by many stochastic factors [36]. In other words, stochastic resilience or ecosystem resilience is the capacity of an ecosystem to overcome disturbances and reorganize to maintain original fundamental state [36–38]. In case of coral reefs, the disturbances vary in terms of time and space with different intensities. As a result, depending on the location and local environmental factors, resilience of a given reef and its coral communities will vary. Resilience-based management has been proposed as a realistic model to predict which sustainability measures can be achieved in coral reefs in the face of ocean warming and acidification and various local disturbances, and this strategy will help us set achievable goals for regional and local-scale management programs [39]. Moreover, beyond the ecosystem level, resilience also covers the overall ability of individuals, populations, or communities to respond positively after disturbance and restore some part of their original state [4]. For example, individual corals can show physiological resilience via survival, sustained growth, reproduction, and/or by shuffling their symbionts towards more thermally-tolerant genera and species (see the review by Carballo-Bolaños et al. in this issue). Coral populations can re-populate by recruiting new individuals, and communities can re-organise ecosystem traits such as productivity, diversity, trophic linkages, and sustained biomass through shifts in species composition [reviewed in 4]. In this review, we adopted the broad definition of resilience [4] to examine the resilience of coral-Symbiodiniaceae associations and coral communities to the long-term disturbances in KNP.

*1.3. Coral-Symbiodiniaceae Associations Play a Key Role in Coral Resilience to Thermal Stress*

Coral-Symbiodiniaceae associations are the crux of coral health and functioning, even though corals are multi-bionts [40]. One of the most important factors of this association is its resilience to seawater temperature anomalies. Coral-Symbiodiniaceae resistance mechanisms to temperature stress depend on the combination of mechanisms involving the coral host and/or its Symbiodiniaceae partners and whether the relationship that the coral host has with Symbiodiniaceae is specific or flexible [41–43]. Corals are known to associate with a wide range of Symbiodiniaceae genera. There are nine genera in Symbiodiniaceae, and each genus has its own characteristics that help corals survive in a wide range of environmental niches [44]. Studies have shown that symbiosis between coral hosts and different Symbiodiniaceae genera contribute to the divergence in their thermal tolerance under different environmental conditions [42,45].

For example, stress-tolerant *Durusdinium trenchii* has proven to be heat-tolerant, whereas *Cladocopium* species are the most sensitive to stress [41,46–48]. *Durusdinium*-associated corals are also known to inhabit reef environments that experience large surface seawater temperature fluctuations [42,49,50] and possess better survival chances under heat-treatment experiments [51]. Symbiodiniaceae diversity thus provides a mechanism for corals to adapt and/or acclimatize to changing environmental conditions. Since environmental factors vary both spatially and temporally, even at a micro-geographic scale, differences in Symbiodiniaceae diversity within a single host or among multiple species are important and contribute to coral resilience.

*1.4. Coral Reefs in Taiwan, with a Focus on Kenting Coral Reefs*

Taiwan, a continental island with several offshore islets, is located at the centre or junction of the Philippine-Japan island arc; the Tropic of Cancer, runs through the middle of Taiwan (Figure 1). Scleractinian coral occurrence and distribution in Taiwan is influenced by sea surface currents and seawater temperatures; it is found in patchy non-reefal coral communities, similar to a high-latitudinal environment, and occupies the Penghu Archipelago and northern, northeastern, and rocky eastern coasts of Taiwan [52–54]. The marine environmental conditions in southern Taiwan—

facing the junction of the Pacific Ocean, Bashi Channel, and South China Sea—are influenced by a branch of the Kuroshio Current (KC) and the South China Sea Surface Current (SCSSC) (Figure 1). Thus, tropical fringing reefs developed along the coast of the Hengchun Peninsula, with about 300 coral species described [52–54].

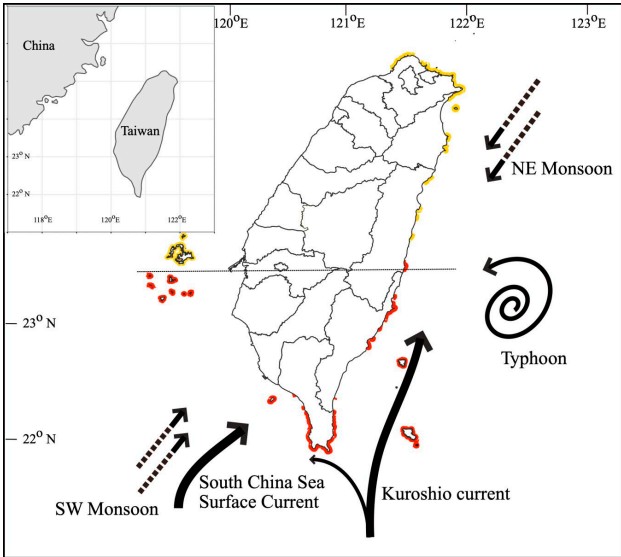

**Figure 1.** The distribution of corals in Taiwan, including non-reefal communities (yellow) and tropical coral reefs (red) divided roughly by the Tropic of Cancer, and the main climate factors, including the Kuroshio Current, typhoons, and northeast and southwest monsoons. The two types of coral communities are divided by the Tropic of Cancer. The information in the map was adapted from [54]. Taiwan map was downloaded from free to use Taiwan Map Store (https://whgis.nlsc.gov.tw/English/0-1Introduction.aspx).

In 1982, Kenting National Park (KNP), the first national park in Taiwan, was established to manage and conserve the uplifted reef landscape as well as the modern coral reef system (Figure 2). In addition, a nuclear power plant (NPP) that started operating in 1985 discharges heated seawater into Nanwan, KNP (Figure 2). Biological surveys and environmental monitoring have been conducted in KNP since the late 1970s to collect baseline data to manage KNP and monitor the environmental impact of NPP during construction and follow-up operations, particularly the thermal stress on the reef adjacent to the heated-water outlet (OL); data collection was interrupted from time to time due to shortages in funding or termination of monitoring projects (Table 1). Nevertheless, this long-term research effort (> 30 years) collected enough data for us to assess spatial and temporal changes in Taiwan's coral reef ecosystems (Figure 2). However, similar to reefs from most of the tropical waters [55–58], the reefs in KNP have been subjected to human disturbances—including overfishing, habitat destruction, and sewage discharge—due to a growing population and poor management [59–62]. Combined with the synergistic effects from climate change, there has been a declining trend in KNP's living coral cover [60,63].

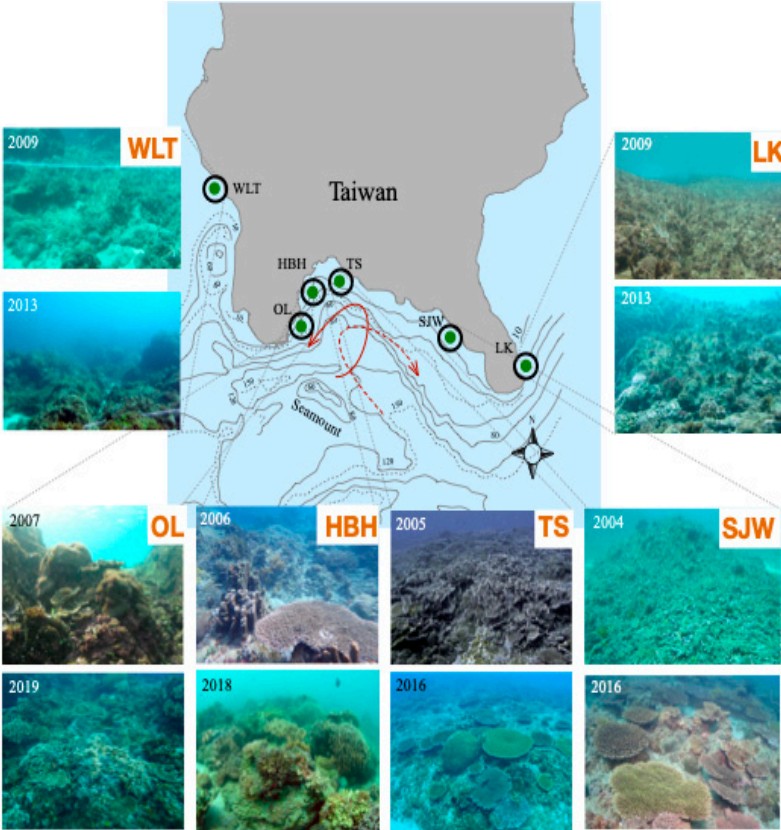

**Figure 2.** Depth contours in Nanwan, Kenting National Park, and coral communities at six sites and two time points, showing the status of coral communities through time. The sites include Wanlitung (WLT) on the west coast of Hengchun Peninsula, NPP Outlet (OL) and Houbihu (HBH) on the west coast of Nanwan, Tiaoshi (TS) and Sianjiaowan (SJW) on the east coast of Nanwan, and Longkeng (LK) on the east coast of Hengchun Peninsula. The solid and dotted arrow line indicates the direction of tidal flow patterns in flows and eddies. Depth countour figure was adapted from [65].

In this paper, we review the long-term studies of (1) environmental settings of KNP coral reefs, focusing on the cooling effect of tide-induced upwelling; (2) spatial and temporal dynamics of benthic communities and how they respond to large-scale disturbances, such as coral bleaching and typhoons; and (3) Symbiodiniaceae diversity and the responses of KNP corals to thermal-induced bleaching. By synthesizing the long-term ecological research data on coral reefs in KNP, we show that, by introducing resilience-based management programs to reduce local stresses, coral distribution along a small geographic range with unique environmental settings and Symbiodiniaceae diversity can be potentially resilient to climate change.

This section can be summarized as; (1) Coral reefs are facing increasing pressure from natural and anthropogenic disturbances and (2) Resilience of corals to such disturbances will depend on local-scale management programs, physiological resilience of individual species, recruitment potential, etc.

**Table 1.** Summary of the historical dataset conducting National Park-wide qualitative survey since 1986.

| Year | Sites | Survey Method | Survey Area per Transect | Number of Replicates at each Site | Survey Depth | Methods | Identification Level | Reference |
|---|---|---|---|---|---|---|---|---|
| 1986–1987, 1998–1999 | Wanlitung (WLT) Hungchai (HC) Leidashih (LDS) Houbihu (HBH) Tiaoshi (TS) Siangjiao Bay (SJW) | Line intercept transect | 10 m | 25 | 3–23 | A transect tape was placed perpendicular to the coast and extended seaward from 3 m depth to the reef edge at 25 m depth. A 10 m metal chain was placed parallel to the transect at 15 m intervals. | Species for corals Total algae | 79–82 |
| 2003–2005 2008–2014 2016, 2018 | Howan (HW) Wanlitung (WLT) Hungchai (HC) Leidashih (LDS) Houbihu (HBH) NPP Inlet (IL) Tiaoshi (TS) Siangjiao Bay (SJW) Longkeng (LK) Jialuoshui (JLS) | 30 m × 0.25 m belt transects except 20 m × 0.25 m in TS | 7.5 m² and 5 m² in TS | 3, except 9 in TS | 5–10 | Three or nine (at Tiaoshi) permanent belt transects were established along depth contours between 5 and 10 m depth at each site. Benthic organisms were quantified using 25 × 25 cm² photo-quadrats.The percent cover of the benthic categories was determined using Coral Count with Excel Extensions software [104], with 30 random points per quadrat. Surveys were conducted between March and May each year. | Species for corals (2003–2005, 2011, 2014); Genus for corals (2008–2010, 2012–2013); Morphology for scleractinian corals (2016); Genus and morphology for scleractinian corals (2018) Macroalgae Turf algae | 62, 85–101 |

## 2. Environmental Settings of Coral Reef in KNP with Focus on Tide-Induced Upwelling

KNP is located on Hengchun Peninsula in southernmost Taiwan and receives seasonal influence from a branch of KC and SCSSC; the park can be divided into three geographic sub-regions, namely the west coast of Hengchun Peninsula facing the Taiwan Strait; the east coast of Hengchun Peninsula adjacent to the Pacific Ocean; and Nanwan, a semi-enclosed bay between two capes facing the Bashi Channel that connects the Pacific Ocean and Taiwan Strait (Figure 1). Distance between the east cape, known as Eluanbi, protruding farther south and the west cape, Maobitou, is about 14 km. The semi-enclosed basin of Nanwan is characterized by a zonally elongated seamount partially blocking the bay mouth. The seamount reaches up to 50 m below the sea surface (Figure 2). Toward the west side of Nanwan, there is almost no continental shelf. On the contrary, the shallow continental shelf shoreward of 80 m isobath is about 4 km wide, with isobaths running more or less parallel to the coastline at the east side of Nanwan. With this unique geomorphologic setting, the deeper portion of Nanwan forms an arc-shaped channel open at both ends between the seamount to the south and landmasses to the north [61–63].

By applying acoustic Doppler current profiler surveys, moored measurements, and numerical modelling, it has been shown that the headlands on either side of the Nanwan generate strong tidally-induced upwelling (TIU) within the bay during each phase of the tide [64,65]. Considerable difference in size between the flood and ebb eddies were observed following the geometry of the region. The entire Nanwan basin is filled by the flood eddy, while the western and central regions are filled by the ebb eddy. Eventually, the upwelling occurs within each eddy, causing two temperature drops per tidal cycle in western and central Nanwan, and only one drop in the eastern part (Figure 2) [64,65].

The significant temperature drops caused by TIU (Figure 3) might not only play an important role in shaping the coral community structure in different sub-regions of KNP, but also have a crucial cooling effect on corals in the Nanwan that resists the rising seawater temperature, both locally and globally. Analysis of in situ hourly temperature data showed a spatial heterogeneity in response to TIU at different sub-regions in KNP (Figure 3). No TIU was detected on the west coast of Hengchun Peninsula; thus, seawater temperature is constantly stable, with small fluctuations through the year (Figures 3a,b). In contrast, within the Nanwan and the east coast of Hengchun Peninsula, seawater temperature responds in concordance, but with different amplitudes, to the TIU, following the moon phases (Figures 3c–j). In west Nanwan, where the OL is located, large amplitudes were observed around new moons, with a maximum temperature fluctuation of 4.08 °C and 4.89 °C at 2 m and 7 m deep in the summer, respectively (Figures 3c–f), providing a significant cooling effect to remove thermal stress on coral reefs near the OL (Table 2). Towards east Nanwan and the corner of east Hengchun Peninsula, the effect of temperature drop by TIU is reduced in the summer, as the model predicted [64,65], but remains relatively strong in the winter (Figures 3g–j).

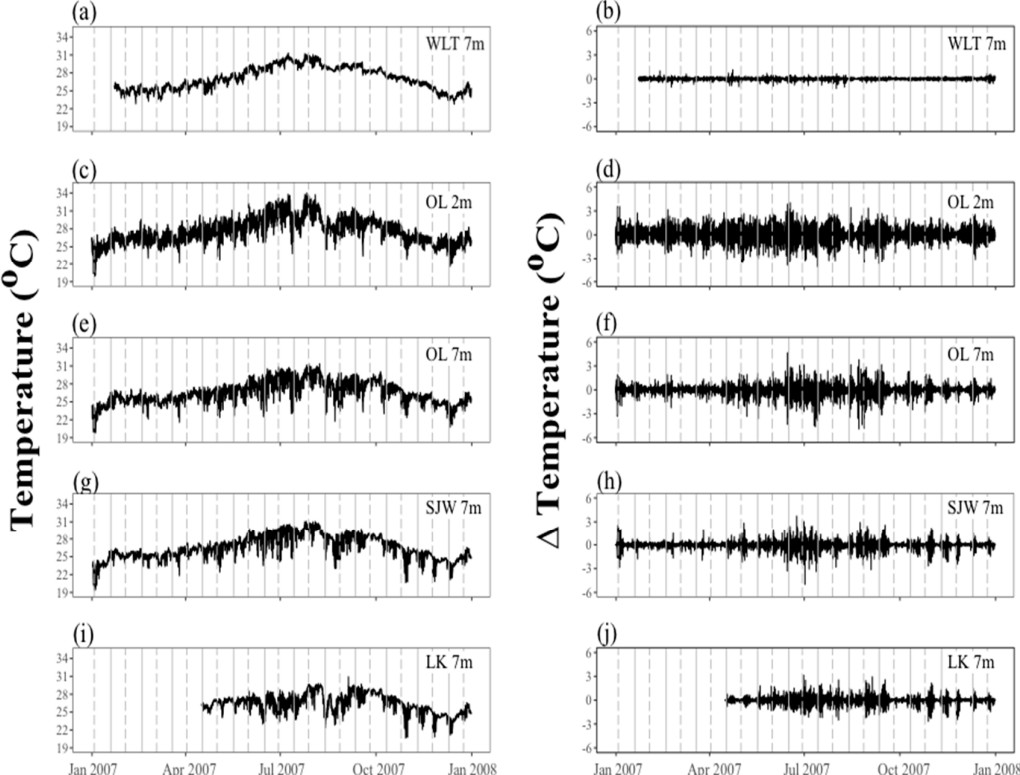

**Figure 3.** The in situ hourly average seawater temperatures and hourly average temperature differences among Wanlitung (WLT) (7 m deep: **a,b**), Outlet (OL) (2 m: **c,d**; 7 m: **e,f**), Sianjiao Bay (SIW) (7 m: **g,h**), and Longkeng (LK) (**i,j**) from January 1, 2007 to January 1, 2008. The vertical dotted and solid lines indicate full moon and new moon, respectively. The seawater temperature was recorded using in situ underwater data loggers (Hobo Pendant® Temperature data logger, Onset Computer Corporation, Bourne, MA, USA). We calculated the hourly average temperature first because temperature were measured at different time interval, such as every 15 min or every 30 min, among sites. Hourly average temperature differences were defined as the difference in hourly average temperatures between any given hour and the hour next. We used hourly instead of daily average temperature difference in order to reveal the temperature difference caused by TIU instead of day cycle.

This section can be summarized as; (1) There is a presence of strong tidally induced upwelling in Nanwan and (2) Significant temperature drops caused by upwelling could possibly play a major role in shaping coral community structure and long-term resilience in different sub-regions of KNP

*2.1. Spatial and Temporal Variability of Coral Communities in Responding to Large-Scale Disturbances*

The coral reefs in KNP have been explored since the latter half of Japanese rule, when efforts began to describe coral species and produce a checklist [66–73]. Efforts to describe reef geomorphology and coral communities began in the 1970s, when KNP was established and the third nuclear power plant was constructed [73–80]. These studies focused on describing the reef morphology, coral and fish diversity, and ecology of planktons and fishes. The coral reefs in KNP are separated by sand channels of various widths and vary in their coral fauna among different locations [summarised in 79]. There is no significant correlation between species diversity (Shannon-Wiener's index, *H'*) per 10 m line intercept transect and depth, except at the northern tip of Nanwan, where the species diversity decreases with depth [79,80]. It was concluded that KNP contains two types of coral communities, one mainly composed of scleractinian corals with a few alcyonaceans (less than

6% in total coverage) in the wave-protected areas, and the other dominated by alcyonaceans in wave-exposed areas (total coverage > 50%) [80].

**Table 2.** Summary of the historical disturbances, typhoons, coral bleaching events, and biological outbreaks in Kenting National Park since 1986. Type I and IV of typhoons referred the direction of typhoon used in Figure 4. Typhoon categories are inside the brackets after typhoon names.

| Category | Name | Year | Note |
|---|---|---|---|
| Typhoon | Peggy (5) | 1986 | Type I |
| Typhoon | Gerald (4) | 1987 | Type I |
| Typhoon | Lynn (5) | 1987 | Type I |
| Typhoon | Herb (5) | 1996 | Type I |
| Typhoon | Chanchu (4) | 2006 | Type IV |
| Typhoon | Morakot (2) | 2009 | Type I |
| Typhoon | Nanmadol (5) | 2011 | Type I |
| Typhoon | Usagi (4) | 2013 | Type I |
| Typhoon | Soudelor (5) | 2015 | Type I |
| Typhoon | Meranti (5) | 2016 | Type I |
| Typhoon | Megi (4) | 2016 | Type I |
| Temperature anomaly | Bleaching | 1998 | Nearly all the colonies on the reefs shallower than 5m in depth in Outlet were bleached. |
| Temperature anomaly | Bleaching | 2002 | Minor, very small scale and local bleaching event were recorded in Wanlitung, Houbihu, and Sianjiaowan |
| Temperature anomaly | Bleaching | 2007 | 50% In Outlet and up to 25% on the West coast of Hengchun Peninsula and Nanwan |
| Temperature anomaly | Bleaching | 2010 | Minor scale on the hallow reef of the NPP OL |
| Temperature anomaly | Bleaching | 2014 | There were around 30% of the corals bleaching in KNP except in the Outlet that 50% and 20% of the corals were bleached on the shallow (shallower than 5m in depth) and deep (10m in depth) reefs. |
| Temperature anomaly | Bleaching | 2016 | Minor scale from Outlet to Nanwan beach |
| Temperature anomaly | Bleaching | 2017 | Minor scale on the West coast of Hengchun Peninsula |
| Ship grounding | Amorgos | 2001 | Limited on the East coast of Hengchun Penunsula, in particular Longkeng |
| Ship grounding | Colombo Queen | 2009 | East coast of Hengchun Peninsula |
| Ship grounding | WO-BUDMO | 2009 | West coast of Hengchun Peninsula |
| Biological outbreak | Sea anemone *Condylactis* sp. | Late 1996–2008 | Limited in the shallow area of Tiaoshi |
| Biological outbreak | Green alga *Codium edule* | Late 1996–2002 | Limited in the shallow area of Tiaoshi with significantly seasonal variation |

Local tourism significantly increased in the 1990s, and long-term ecological research (LTER) was introduced to monitor the impact of local human disturbances—such as overfishing, habitat destruction, and pollution—on KNP coral reefs [59,60,62]. Observations have shown that variations in large-scale physical disturbances, such as typhoons, in different sub-regions had more of an impact on environmental and biological processes than human disturbance, and resulted in spatial heterogeneity of coral communities in KNP [78]. Analysis showed that typhoons in Taiwan took four types of routes between 1911 and 2018: 26.3% traveled north-westward (Type I), 12.3% traveled northward along the east coast (Type II), 10.26% traveled south-eastward (Type III), and 6.84% traveled northward along the west coast (Type IV) (Figure 4). Interestingly, historic records suggest that Type I typhoons are the major contributor of large-scale disturbances that shape coral communities in KNP (Table 2).

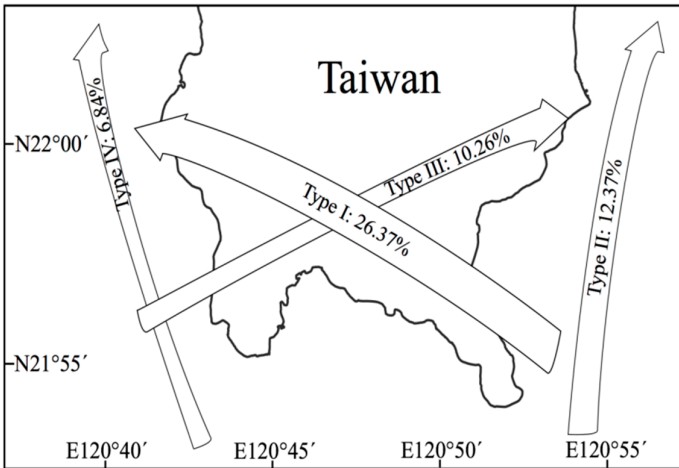

**Figure 4.** Common trajectories that typhoons took through southern Taiwan between 1911 and 2018. Type I and III indicate a typhoon running north-westwards and north-eastwards, respectively. The other two indicate a typhoon running northwards along the east (type II) and west coasts (type IV). The data were collected from the Typhoon Database, Central Weather Bureau, Taiwan.

LTER data of 5 to 10 m depths showed a declining trend in mean living coral cover (LCC) in KNP, from 48.56% in 1986 to 29.33% in 2018 (Figure 5a), mainly due to the synergistic effects from multiple disturbances (Table 2). Three time-intervals—1986–2000, 2000–2006, 2006–present—shed light on coral community dynamics. First, although no data is available between 1987 and 1997, typhoons in 1986, 1987, and 1996 and the 1998 mass bleaching event were proposed to account for LCC decreasing to 36% by 1999 [81,82], and as a result of local pollution and habitat destruction, was followed by outbreaks of the macroalgae *Codium* spp. [83] and sea anemone *Condylactis* sp. [84] and see references in Table 1 [62,79–82,85–101]. Second, between 2000 and 2006, the LCC returned to a level similar to that of 1986 (> 45%) in 2003, 2004, and 2005 (Figure 5a) due to a lack of major typhoons or bleaching (except a minor one in 2002) between 1999 and 2005 [62,101].

Interestingly, dominant species that were recovered and found to contribute to the increasing LCC during this period were significantly different from those in 1986; for example, in Wanlitung, a LTER site on the west coast of Hengchun Peninsula, the LCC was composed of *Montipora*, *Heliopora*, and *Poritidae* during this period, whereas *Acropora* was the dominant coral genus in 1986 [62,101]. Third, intense disturbances by typhoons (Table 2) caused LCC in KNP to decline in 2006 and stay low until 2016 (Figure 5a). For example, Typhoon Morakot in 2009, the deadliest one in the recorded history (although recorded as category 2), stayed on top of Taiwan for 2 days, causing flooding and big waves that brought the LCC down to 21.07% in 2010 [102,103].

Although the LCC recovered by 2016 to 43.86%, typhoons Meranti and Megi (ranked category 5 and 4, respectively, Table 2), directly hit KNP in September 2016 and combined with minor coral bleaching in 2017, again caused a dramatic decline of LCC down to 29.33% in 2018 [103].

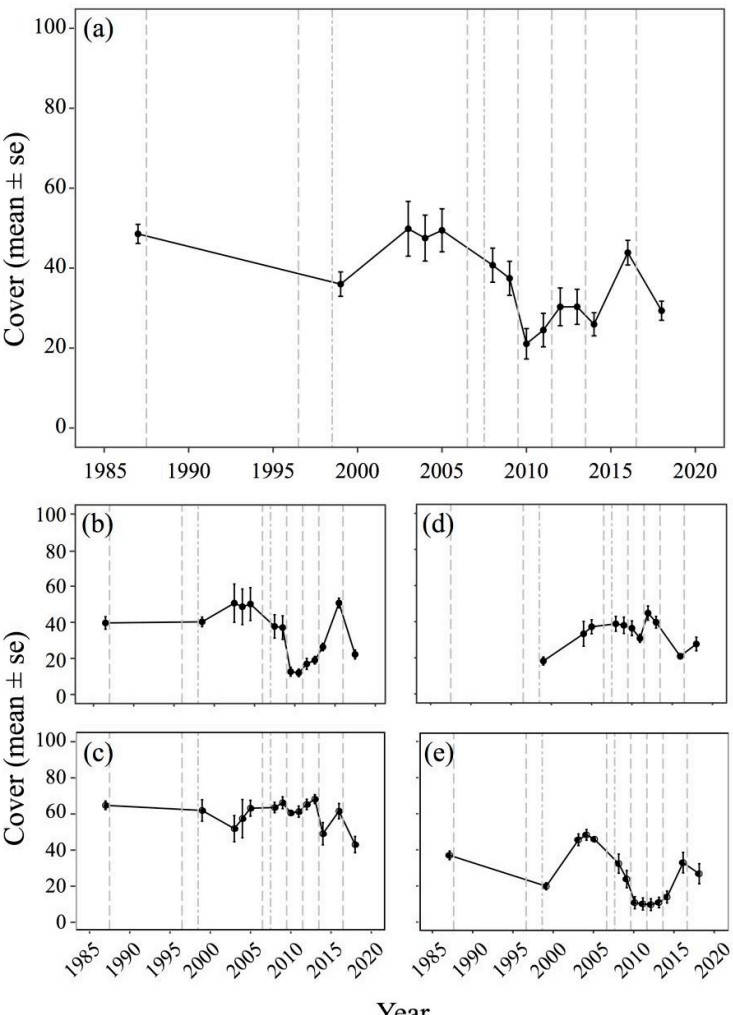

**Figure 5.** Spatial and temporal long-term trend of average living total coral cover (LCC, per transect ± standard error) in KNP from 1985 to the present. (**a**) the average LCC in the Kenting National Park scale, (**b**) the west, and (**d**) the east coast of Hengchun Peninsula; (**c**) the west, and (**e**) the east coast of Nanwan. The wide and narrow dashed lines indicate typhoons and bleaching events in KNP. The data were collected from the dataset listed in Table 1. The detailed survey methods of the data used in this figure are listed in Table 1. In order to compare the LCC at similar depth, only the transects laid out between 5 and 10m in depth at each site from 1986 to 1999 [79–82], were used to generate the figure, combined with the data collected after 2003, including data from a PhD thesis [79], one article published in the proceeding of the 6th International Coral Reef Symposium [80] and two local journal articles written in Mandarin with English abstract [81,82]. The data of LCC after 2003 were collected by the co-first author C.Y.K. and the corresponding author C.A.C. Different parts of this dataset have been publised as a Master's thesis [101], in local project reports [85–100] and journal articles [62,102].

Spatial variability in the response of coral reefs to disturbances was observed among sub-regions in KNP (Figures 5b–e) by analysing the coral cover using the method described in [104]. The coral reef at the west coast of Hengchun Peninsula (Figure 5b) and east coast of Nanwan (Figure 5c) showed a similar trend of LCC dynamics of the overall KNP, but the latter showed no LCC decline between 1986 and 2000. The east coast of Hengchun Peninsula showed an increasing trend at the time interval

between 2000 and 2006, declined due to the impact of typhoon Morakot in 2009, and reached its highest LCC of 44.77% in 2012 before declining again between 2012 and 2016 and, finally recovering by 2018 (Figure 5d). The west coast of Nanwan has maintained a higher LCC (> 40%) than other subregions in the last 36 years (Figure 5e) due to the lack of direct impact from typhoons.

Although typhoons have played a notable role of causing the decline in LCC in KNP, it seems they also have benefited the coral community in KNP. The positive effect of typhoons in terms of cooling of the sea surface temperature and breaking down accumulated heat stress in summer by mixing the heated surface water with cooler water from deeper areas [105,106] has resulted in reduced or no coral bleaching. The best example of this in the KNP is NPP Outlet (OL) located at the west coast of Nanwan. This site is most protected from storm surges and at the same time most exposed to the thermal stress caused by the nuclear power plant discharge. In September 2009, while the storm surges of typhoon Morakot did not cause a significant damage on the LCC of the reefs in NPP Outlet (OL), compared to the reefs on the east coast of Nanwan (Figure 5e), it did reduce the temperature of the constantly heated sea surface water at the shallow part of the OL by > 6 °C (Figure 6). The cooling effect created an environmental condition of temperature equal to winter and lasted for 3 days after the typhoon passed.

The coral communities in the reef adjacent to the nuclear power plant outlet (OL) are protected by storm surges and cooled down by TIU and typhoons, however, the local rising and variable sea surface temperatures [107–110] has resulted in shifting dominant coral species (Figure 6) due to multiple coral bleaching events over time (Table 2). Also at NPP OL, the warm discharged water is trapped in the shallow waters (up to 4 m deep) and flows southwestward in Nanwan because of a near-shore current and tides [111], resulting in a 2.0–3.0 °C higher summer average seawater temperature than at other coral reef sites in Kenting [107,112,113]. Comparing the living coral assemblage in 1986, 1995, 2010, and 2019 in shallow water (3 m) at OL, there was a sharp change in coral genus composition in 1995 (Figure 6a). While *Acropora* dominated in 1986 (31.58% of relative LCC) and 1995 (59.42%), *Galaxea* replaced it and became dominant in 2010 (31.64%) and 2019 (21.11%). *Montipora* remained relatively constant throughout the monitoring period, and *Seriatopora* and *Stylophora* were completely absent at 3 m at OL after 2010 [107]. In addition, coral genus composition remained similar in 1986, 1995, 2010, and 2019 in deep water (7 m), although their relative abundance fluctuated through time (Figure 6b).

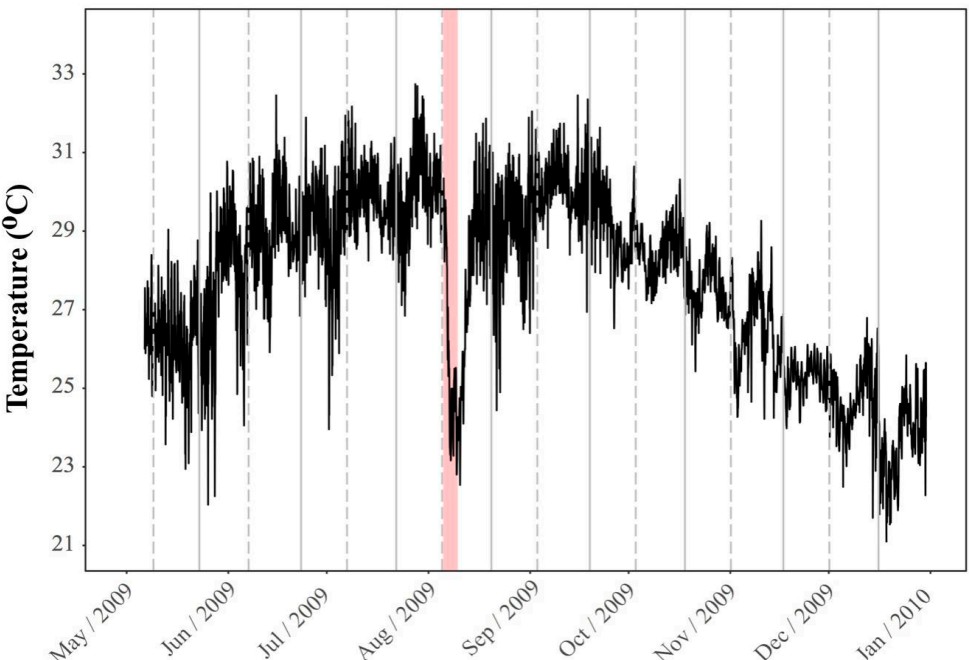

**Figure 6.** The in situ hourly average seawater temperatures at 2m depth in Outlet (OL) from June 6, 2009 to December 31, 2009. The vertical dotted and solid lines indicate full moon and new moon,

respectively. The pink area indicates the period, from 5th August. 20:30 to 10th August 5:30 2009, the sea warning for typhoon Morakot, the deadliest one in the history of Taiwan, was issued by the Central Weather Bureau.

However, the response of coral communities to typhoons are varied and has resulted in spatial variation in long term changes of LCC among subregions in KNP. In the nuclear power plant outlet (OL), the reef most protected from typhoons, local, small scale variation of temperature has caused multiple bleaching events and resulted in the loss of temperature sensitive taxa.

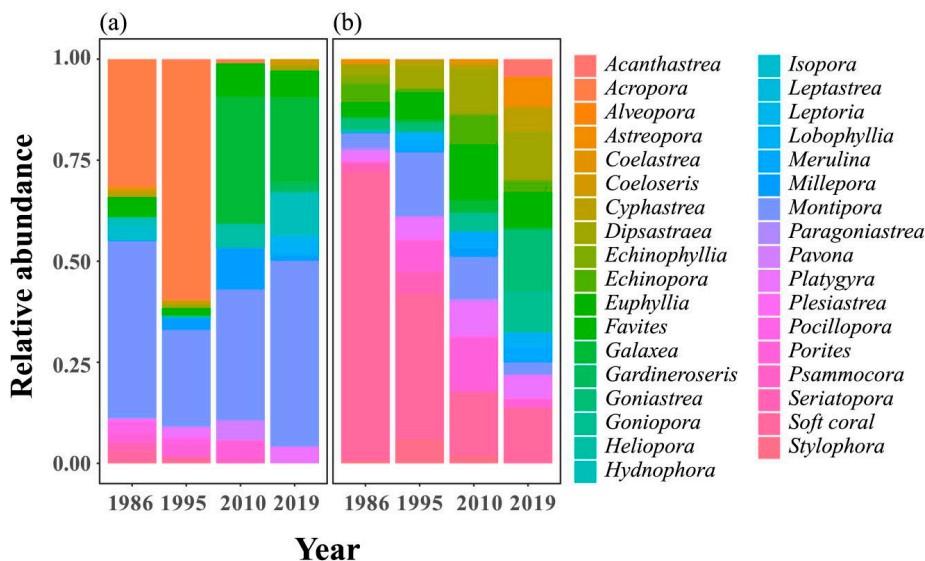

**Figure 7.** The relative abundance of coral genera at 3 m (shallow) and 7 m (deep) in NPP OL in 1986, 1995, 2010, and 2019. This figure was redrawn using data from Figure 7 in [109] and combined with a survey conducted in 2019.

This section can be summarized as; (1) The effect of typhoons and bleaching has made notable impact on environmental and biological processes resulting in spatial heterogeneity of coral communities and (2) Response of coral communities to typhoons are varied and has resulted in spatial variation in long term changes of LCC among subregions in KNP. In the nuclear power plant outlet (OL), the reef most protected from typhoons, local, small scale variation of temperature has caused multiple bleaching events and resulted in the loss of temperature sensitive taxa.

### 2.2. Symbiont Community Dynamics over Space and Time in KNP

Coral-associated Symbiodiniaceae from different locations in KNP have been analysed since 1997 [107–110,114–116]. Our studies on Symbiodiniaceae diversity in KNP over the past 24 years demonstrate fine-scale, micro-geographic, temporal, and species-specific associations among genera in addition to/other than coral-associated *Cladocopium* spp. In particular, the ability of corals to associate with multiple Symbiodiniaceae genera and change (i.e., shuffle) between stress-resistant and stress-sensitive genera/species depending on environmental conditions [46,47,117–122] is a critical requirement of their resilience towards stress. At the community level, *Cladocopium* spp. were the dominant Symbiodiniaceae associated with corals in KNP (previously Clade C), as they are elsewhere in the Pacific and South China Sea [123–125]. No matter the molecular technique applied to phylotype Symbiodiniaceae, the results consistently shows that *Cladocopium* sp. is dominant and co-occurs with *Durusdinium* sp. (previously Clade D).

**Table 3.** Coral-Symbiodiniaceae associations in Kenting National Park through time analyzed using more sensitive over time. Until 2010 all data are shown by old taxonomy used for Symbiodiniaceae. C = *Cladocopium* spp. (previously clade C), D = *Durusdinium* spp. (previously clade D).

| Year | Host Species (Family/Genus) | Symbiodiniaceae Clade/Type/Genera/Species | Study Sites in KNP | Genetic Method for ID | Reference |
|---|---|---|---|---|---|
| 1997–2001 | *Acropora* | C3, C1, D1, D2, | | srDNA- RFLP | [114] |
| | *Montipora* | C1 | | | |
| | Pocilloporidae | C1, C2 | | | |
| | Euphyllidae | C1, D1 | | | |
| | Poritidae | C1 | | | |
| | Siderastreidae | C1 | | | |
| | Agariciidae | C1 | | | |
| | Oculinidae | C3 | | | |
| | Merulinidae | C1 | | | |
| | Faviidae | C1 | | | |
| 2000–2001 | *Isopora palifera* | C, D | Tantzei Bay | srDNA- RFLP | [108] |
| 2006–2009 | *Isopora palifera* | C3, D1a | Tantzei Bay, Maobitou, | srDNA- RFLP | |
| | | | Siatanzai, NPP-OL, | ITS2-DGGE | |
| | | | Shiaowan, Shinjaowan | | |
| | | | Longken | | |
| 2009 | *Platygyra verweyi* | C3, D1a | Lidashih, Siatanzai | ITS2-DGGE | [107] |
| | | | Maobitou, NPP-OL | ITS1-qPCR | |
| | | | Wanlitung, Hongcha | | |
| | | | NPP-IL, Tiaoshi, | | |
| | | | Tantzei Bay, Longken | | |
| 2009–2010 | *Acanthastrea* | C1, D1a | Houbihu, NPP-OL | srDNA- RFLP | [109] |
| | *Acropora* | C21a, C3, D1a | Shinjiaowan, NPP-IL | ITS2-DGGE | |
| | *Cyphastrea* | C3, D1a | Wanlitung, Tiaoshi | ITS1-qPCR | |
| | *Favia* | C3, D1a | Tantzei Bay, Longken | | |
| | *Favites* | C3, D1a | | | |
| | *Galaxea* | C1, D1a | | | |
| | *Goniastrea* | C1, D1a | | | |
| | *Isopora* | C3, D1a | | | |
| | *Leptastrea* | D1a | | | |
| | *Leptoria* | C1, D1a | | | |

| | | | | | |
|---|---|---|---|---|---|
| | *Montastrea* | C1, C3, D1a | | | |
| | *Montipora* | C15, D1a | | | |
| | *Pavona* | C1, D1a | | | |
| | *Platygyra* | C3, D1a | | | |
| | *Pocillopora* | C3, D1a | | | |
| | *Porites* | C15, D1a | | | |
| | *Seriatopora* | C1 | | | |
| | *Stylophora* | C1 | | | |
| **2016-2017** | *Leptoria phyrgia* | *Durusdinium glynii* *Durusdinium trenchii* *Cladocopium* C3w *Cladocopium* C21a *Cladocopium* sp. | Wanlitung NPP-OL | ITS2-DGGE ITS1-qPCR | [110] |
| **2019** | *Leptoria phyrgia* | *Durusdinium glynii* *Durusdinium trenchii* *Durusdinium* D1.6, D17, D2, D5, D6 *Cladocopicum* C116, C15.7, C21a, C2r, C3.1 C3.8, C3b, C3d, C3e, C3s, C50 | Wanlitung NPP-OL | ITS2 Amplicon | [115] |

RFLP = Restriction Fragment Length Polymorphism, DGGE = Denaturing Gradient Gel Electrophoresis, qPCR = Quantitative Real Time Polymerase Chain Reaction

Fine-scale techniques such as ITS2-DGGE, used from 2009 onwards, revealed a fine-scale variation in associated species within each genus; for example, *Cladocopium* C1, C3 and *Durusdinium trenchii* (previously D1a) have all been shown to be dominant. However, recent use of up-to-date NGS amplicon sequencing has revealed more diversity within genera *Cladocopium* and *Durusdinium* (Table 3). In addition, a study published in 2014 [109] showed depth and species-related differences in the coral-associated Symbiodiniaceae in KNP. Samples collected from 16 genera from eight locations and two depths in KNP revealed some interesting trends (Figure 8). *Cladocopium* spp. were more dominant in deeper than shallow water, especially in the corals occurring near OL and in Nanwan.

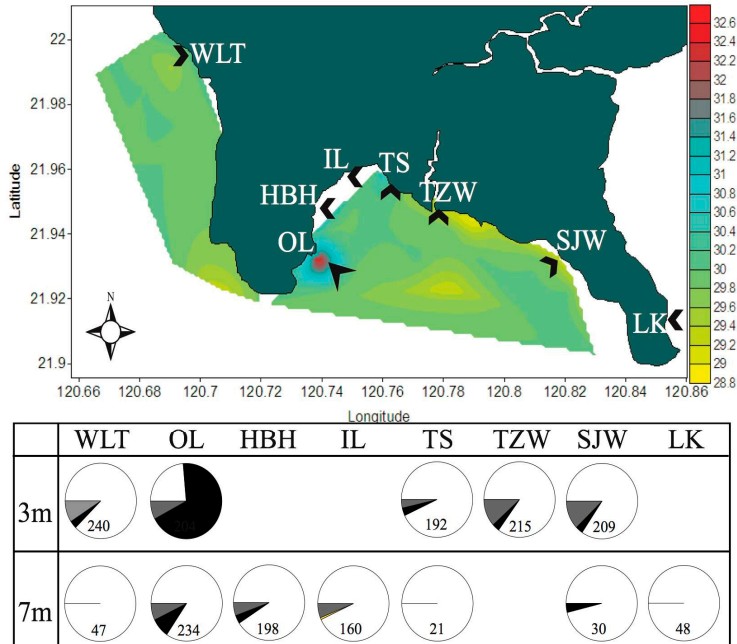

**Figure 8.** Symbiont diversity associated with corals at different locations—Wanlitung (WLT), NPP outlet (OL), Houbihu (HBH), NPP Inlet (IL), Taioshi (TS), Tanziwan (TZW), Sanjiawan (SJW), and Longken (LK)—and from the reef near the nuclear power plant outlet (OL) in Kenting National Park. Samples were collected in 2009 and 2010 from 3 m and 7 m depths. Pie-Charts were drawn using data from previous publications [109]. Analysis of samples collected was done using srDNA-RFLP, ITS2-DGGE and ITS1-qPCR (see Table 3). White = *Cladocopium* sp. (previously clade C), Black = *Durusdinium* sp. (previously clade D), and Grey = co-occurrence of *Cladocopium* sp. and *Durusdinium* sp. The values inside the pie-charts are total sample numbers for each location.

For example, *Isopora palifera* samples in the Tanziwan (TZW) population were dominated by *Durusdinium* spp. in 2001 (three years after the 1998 mass coral bleaching event), *Cladocopium* spp. in 2005, co-dominant by two Symbiodiniaceae genera in 2009, and returned to *Durusdinium* dominance in 2015 (Figure 8). This is concordant with previous studies that show that the occurrence of multiple Symbiodiniaceae genera at low concentrations densities might lead to either shuffling or switching to beneficial Symbiodiniaceae genera over time [121,122,126]; in some cases, the coral host may revert back to its original composition of either a single dominant Symbiodiniaceae species or multiple dominant species/genera [108,122]. In addition to temporal changes, our study also indicated a very efficient spatial difference in Symbiodiniaceae associated with *I. palifera*. Samples collected from different locations in KNP showed [108] that corals at a site next to the OL were exclusively associated with or dominated by *D. trenchii/glynii,* and those at sites away from the OL were associated with *Cladocopium* C3 (Figure 9). Such a micro-geographic difference in Symbiodiniaceae composition is due to the presence of the OL, which has released hot water onto to the coral reefs for over 35 years. Hence, corals near the OL have adapted/acclimatized to associate with *Durusdinium* spp. that are generally stress- as well as temperature-tolerant. Similar spatial differences in the association with Symbiodiniaceae was found in the coral *Platygyra verweyi* and *Leptoria phygria* [108,110]. Nonetheless, there are also cases, irrespective of environmental perturbations, in which the host maintains stable symbiosis with a particular Symbiodiniaceae genus [123,127,128].

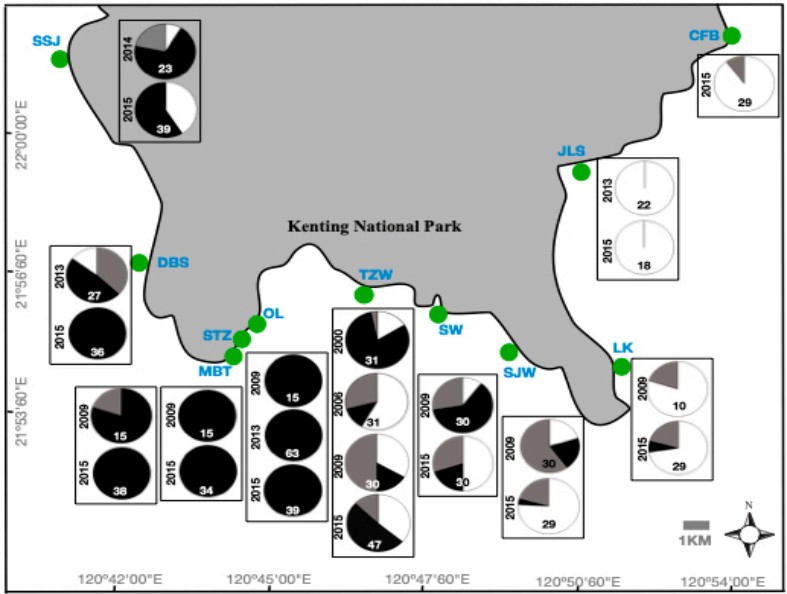

**Figure 9.** Spatial and temporal variation in *Cloadopoium* sp. and *Durusdinium* sp. data in the coral *Isopora palifera* from different locations—Siashuijue (SSJ), Dingnaisha (DBS), Maobitou (MBT), Siatanzai (STZ), nuclear power plant outlet (OL), Tanziwan (TZW), Shiaowan (SW), Sanjiawan (SJW), Longken (LK), Jialeshui (JLS), and Chufengbi (CFB) in Kenting National Park. The data were collected from 2000 to 2015 are adjacent to the pie-charts. Pie-Charts were drawn using data from previous publications [108,113]. Analysis of samples collected in 2000 was done using srDNA-RFLP and all the samples obtained between 2006–2015 were analysed by srDNA-RFLP and ITS2-DGGE (see Table 3) White = *Cladocopium* sp. (previously clade C), Black = *Durusdinium* sp. (previously clade D), and Grey = co-occuurence of *Cladocopium* sp. and *Durusdinium* sp. The values inside the pie-charts are total sample numbers for every location.

This section can be summarized as; (1) Dominant Symbiodiniaceae associated with corals in the KNP is Cladocopium sp. However, majority of coral species in the shallows of NPP OL are associated with Durusdinium sp. Shuffling of Symbiodiniaceae in corals is almost non-existent.

## 3. Discussion and Conclusion

There is a serious concern that coral reefs will almost entirely disappear by 2050 if average ocean temperatures increase by 2 °C, with just 10–30% of existing reefs surviving if ocean temperatures increase by 1.5 °C [19]. The "mission-impossible" goal is to drastically reduce $CO_2$ emissions to net zero and maintain an only 1.5 °C temperature increase; this might lead to 10% of current reefs surviving after 2050. Some global efforts, such as the 50 Reefs Initiative, have used Modern Portfolio Theory (MPT) to identify coral reef locations that represent imperative conservation investments and to ensure their survival; the goal of these efforts is to prepare these areas for repopulation once the climate has been stabilized [129]. If 10%–30% of existing reefs will indeed survive if ocean temperatures increase by only 1.5 °C, an important question is what to do if a reef does not pass the MPT criteria, but does possess certain localized environmental settings, reef topologies, and coral species that do or could resist the impacts of climate change. Herein we argue that the coral reef in KNP could have this great potential to resist the impacts of climate change and deserves novel conservation efforts to ensure its contribution to the coral reef resilience not only in Taiwan but also in the West Pacific.

First, the KNP coral reefs probably receive a constant supply of coral larvae from the south to replenish the reefs after disturbances, and also serve as a "stepping stone" to connect to the reefs or coral communities at high latitudes with currents such as the Kuroshio Current (KC) and the South China Sea Surface Current (SCSSC). Southern Taiwan is bordered by Luzon Island (the Philippines),

the north boundary of the "Coral Triangle" (CT). It is argued that KNP has a relatively high scleractinian and reef-associated species diversity because of its connection to the CT [52–54]. Preliminary studies on the genetic connectivity of coral reef fishes in the West Pacific and South China Sea support this scenario [130,131]. Further research on the genetic connectivity of scleractinian corals using high-resolution molecular markers could help elucidate the resilience role of the KNP reef in the region.

Second, many typhoons in the north-western Pacific pass through Taiwan; thus, typhoons might play both negative and positive roles in shaping the coral community structure at different sub-regions in KNP. Of the four types of typhoons recorded (Figure 4), type I (Table 2)—created by a southeast-northwest vortex—contributes most to the temporal dynamics and spatial heterogeneity of coral communities in different sub-regions of KNP (Figure 5). While there is always some mechanical damage, KNP coral reefs can also benefit from typhoons during the warm summer months. As ocean surface waters become warmer during the summer, corals often experience thermal stress. Typhoons can relieve thermal stress by (1) absorbing energy from surface waters through the transfer of latent heat; (2) inducing local upwelling, bringing deeper, cooler water to the surface; and (3) creating clouds of typhoons to shade the ocean surface from solar heating, allowing the water to cool and reducing light stress. Although the projected impacts of climate change on typhoons remain debatable, monitoring the physical damages caused by typhoons and their joint effect with thermal-induced coral bleaching will be crucial for us to develop a management plan to improve coral reef resilience in KNP.

Third, upwelling has been proposed to be a cooling mechanism to protect coral reefs against bleaching by reducing seawater temperatures or creating fluctuating thermal environments that induce corals to build thermal resistance over time [132–135]. However, it has also been suggested that upwelling areas do not always guarantee refuge for coral reefs in a warming ocean unless the thermal threat and upwelling coincide [136]. Some large-scale seasonal upwelling with cold, nutrient-rich, and naturally acidic subsurface water—such as the upwelling in Gulf of Panama and Papagayo upwelling of Costa Rica, in the tropical eastern Pacific—indeed hinders the development of coral reefs [137–139]. In KNP, upwelling is small-scale, localized, and induced by tides flowing from east to west Nanwan that create a temperature difference within the bay of Nanwan and two sides of Hengchun Peninsula, where the water is cooler in the east and warmer in the west (Table 4).

**Table 4.** Characteristics of the in situ temperature record at the five sites from January 1, 2007 to January 1, 2008.

| Subregions | Site | Depth (m) | Yearly Mean Temperature | Max SST | Min SST | Max. Variability Within 2 Hours |
|---|---|---|---|---|---|---|
| West Hengchun Peninsula | Wanlitung | 7 | 27.090 | 31.33 (July) | 22.77 (Dec.) | 3.15 (July) |
| West Nawan | Outlet | 2 | 27.580 | 34.02 (July) | 19.91 (Jan.) | 4.08 (July) |
| West Nawan | Outlet | 7 | 26.620 | 31.36 (Aug.) | 18.70 (Jan.) | 4.90 (Aug.) |
| East Nawan | Sianjiao Bay | 7 | 26.580 | 31.09 (July) | 17.82 (Jan.) | 4.99 (July) |
| East Hengchun Peninsula | Longkeng | 7 | 26.460 | 30.93 (Sep.) | 20.63 (Oct.) | 3.13 (June) |

In addition, the upwelling helps reduce the thermal stress, particularly in the reef adjacent to the nuclear power plant (OL), by creating significant temperature drops during spring tides in the summer (Table 4) and fluctuating thermal environments that induce corals to build thermal resistance (Figure 7,8). These positive effects ensure that KNP remains a refuge for coral reefs to survive in a warming ocean. Monitoring whether the tide-induced upwelling will be enhanced or hindered by the rising background seawater temperature in the region should be considered as a research priority in KNP.

Fourth, Symbiodiniaceae play a crucial role in bleaching tolerance. Many species or genera of Symbiodiniaceae have been identified [44], and different genera display varying thermal, and therefore bleaching, resistances. It has been suggested that, by associating with or shuffling the symbiont community towards making thermal-tolerant Symbiodiniaceae, such as *Durusdinium* spp, dominant, corals can increase their thermal tolerance by 1.0 °C–1.5 °C [118,119]. Our long-term monitoring of symbiont community diversity shows that corals constantly exposed to warming and fluctuating thermal environments (OL) or constantly higher seawater temperatures (west coast of Hengchun Peninsula) have a dominance of *Durusdinium* spp, whereas the same species located on the cooler east coast of Hengchun Peninsula are increasingly associated with *Cladocopium* spp. (Figure 7). This subregional difference in symbiont community is concordant with the influence of TIU that pumps cooler water from Eluanbi (east cape), protruding towards Maobitou (east cape), creating two temperature drops per tidal cycle in western and central Nanwan and one drop in the eastern part, but no having impact on the west coast of the peninsula. The TIU affects the seawater temperature in KNP and not only drives sub-regional variability in symbiont communities, but also provides the signal for corals to shuffle their symbionts in response to seasonally fluctuating seawater temperature; this is not, however, true for corals in shallow water (< 3 m) of the reef adjacent to the nuclear power plant OL, which are associated dominantly with *Durusdinium* spp. and do not show sign of shuffling [107–109]. In reciprocal transplantation experiments (RTE), corals from WLT to OL did not survive under a prolonged seawater temperature anomaly, even though they showed signs of shuffling from *Cladocopium* to *Durusdinium* dominance [22]. These results imply that corals in the shallow water of the OL reef already live at the ceiling of thermal tolerance, and future climate change trends might be untenable for those corals [110].

Despite these environmental, ecological, and biological characteristics, adaptive management strategies such as implementing sewage treatment systems, banning the serving of herbivorous fishes in restaurants, and promoting eco-friendly tourism and public awareness in recent years has aided in the resilience of coral reefs in KNP. Present and future adaptive management in accordance with the framework of resilience-based management [39] might help sustain coral reef resilience in Kenting National under the impacts of climate change.

This section can be summarized as; Coral reefs in a small geographical range with unique environmental settings and ecological characteristics, such as the KNP reef, are resilient to bleaching and deserve novel conservation efforts. Thus, conservation efforts that use resilience-based management programs to reduce local stresses and meet the challenge of climate change is urgently needed.

**Author Contributions:** Conceptualization, CAC, JTW, PJM, SK and CYK; Methodology, CAC, SK, CYK, RC-B, YYH.; Formal analysis, CYK, SK, RC-B, YYH; Investigation, CAC, CYK, SK.; Resources, CAC, JTW, PJM; Data curation, SK, CYK; Writing—original draft preparation, SK, CAC, CYK; Writing—review and editing, SK, CAC, CYK.; Funding acquisition, CAC, JTW, PJM.

**Funding:** This research was funded by Academia Sinica for Life Science Research (no. 4010) and Academia Sinica Thematic Grant- Green Island (no. AS-TP-108-LM14) to CAC.

**Acknowledgments:** Many thanks LM Chou and D Huang, the subject editors of this coral reef resilience special issue, for inviting us to contribute to this long-term ecological research (LTER) data project on Kenting National Park, Taiwan. We thank all the students, research assistants, and postdoctoral fellows over the past two decades that helped collect data for this LTER. SK and CYK are supported by Academia Sinica postdoctoral fellowships. This work was funded by the Ministry of Science and Technology, Kenting National Park, and Taiwan Power Company to PJM, JTW, and CAC. We would like to thank Noah Last of Third Draft Editing for his English language editing.

**Conflicts of Interest:** The authors declare no conflict of interest and the funders had no role in the design of the study; in the collection, analyses, or interpretation of data; in the writing of the manuscript, or in the decision to publish the results.

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
