# Peer review of "Coral Reef Resilience in Taiwan: Lessons from Long-Term Ecological Research on the Coral Reefs of Kenting National Park (Taiwan)"

_jmse, doi:10.3390/jmse7110388_

Round 1

Reviewer 1 Report

This is an interesting and comprehensive paper. Given the wide range and detail of information available I think the paper cold benefit from some summary sentences at the end of each section that draws out the key messages to the reader.

The disturbances to the reef system vary in scale (over both time and space, as well as intensity) and how this specifically impacts on reef resilience and explains the nature of their findings could be addressed.

A summary table showing the principle conclusions of the study in the context of type of disturbance would be useful.

The discussion has a degree of repetition with the introductory text - although it is important to refer to the particular condition that is being addressed (e.g. upwelling) it is not always clear what the conclusions are that the authors are drawing in the the context of the specific findings of the study.

The potential for adaptive management strategies to assist reef resilience is treated briefly in the last paragraph and as a conclusion to the paper should be addressed in more detail with some specific recommendations based on the results - especially as a conclusion with stronger reference to previous work on adaptive management in the context of reefs. 

Author Response

Response to Reviewer 1 Comments

Reviewer 1

This is an interesting and comprehensive paper. Given the wide range and detail of information available I think the paper cold benefit from some summary sentences at the end of each section that draws out the key messages to the reader.

Authors Response: Thank you for the suggestion, we have now included summary sentences at the end of each section in the revised manuscript

The disturbances to the reef system vary in scale (over both time and space, as well as intensity) and how this specifically impacts on reef resilience and explains the nature of their findings could be addressed.

Authors Response:We have revised the manuscript taking into consideration the suggestion form the reviewer. Also including summary sentences at the end of each section to make the important points clear.

A summary table showing the principle conclusions of the study in the context of type of disturbance would be useful.

Authors Response: Thank you, since we have included summary sentences at the end of each section, which also includes disturbances, it will be a repeat to make the table.

The discussion has a degree of repetition with the introductory text - although it is important to refer to the particular condition that is being addressed (e.g. upwelling) it is not always clear what the conclusions are that the authors are drawing in the the context of the specific findings of the study.

Authors Response: The repetitions has been removed form the beginning of discussion. Actually the Introduction part ends at Section 1. From section 2 we describe how corals reefs have been responding to environmental settings in the KNP and introduce the effect of upwelling in Section 2, and discuss in detail about it in the Discussion part

The potential for adaptive management strategies to assist reef resilience is treated briefly in the last paragraph and as a conclusion to the paper should be addressed in more detail with some specific recommendations based on the results - especially as a conclusion with stronger reference to previous work on adaptive management in the context of reefs. 

Authors Response: Adding a summary section at the end of discussion section to make the point about management. This manuscript is already very long, and we understand that more detailed explanation will help. But, both in the introduction and discussion we make it clear that given the nature of disturbances and changes in the coral reef ecosystem in the KNP, management policies and framework recommended in Anthony et al 2016 need to be followed. 

Reviewer 2 Report

see file attached

Author Response

Response to Reviewer 2 Comments

Reviewer 2

The ms by Shashank Keshavmurthy et al. reports on a long-term research carried out in a National Park in Taiwan and provides interesting information about the coral reefs of this region.

However, the paper deals particularly with the Coral-Symbiodiniaceae associations and their identification. The title is therefore misleading: please change the title stressing this aspect. As I am not an expert on coral symbiosis, I would recommend that the ms is read by a specialist on coral zooxantellae.

Authors Response: Dear Reviewer, thank you for your suggestion about the title, but we beg to defer. This is a review manuscript and it does not deal particularly with Coral-Symbiodiniaceae associations and their identification. It is one part of this review manuscript. There is detailed discussion about coral community shift, influence of environmental parameters and discussion as to how environmental settings have influenced coral reef in the Kenting National Park

The paper makes an ample use of the resilience concept. It is introduced making reference to the classical Holling’s paper, where the term was applied to ecosystems. The Holling’s concept has been reviewed (e.g., Montefalcone et al., 2011. In: Treatise on estuarine and coastal science, Vol. 10, pp. 49-70. Elsevier, DOI: 10.1016/B978-0-12-374711-2.01003-2, and references therein). Trough the paper, however, the term resilience is applied to either corals (which may be appropriate due to the focus of the ms) or coral reefs, which means an important shift in ecological complexity. A useful reading about resilience viewed at different integration levels in coral reefs may be: Bianchi et al., 2017. In: Marine animal forests: the ecology of benthic biodiversity hotspots, pp. 1241-1269. Springer,

doi.org/10.1007/978-3-319-21012-4_35 (and references therein).

Authors Response: Thank you for the suggestions, those references have been incorporated in the revised manuscript

In the ms it is difficult to catch which are original data and which is literature information. Providing some detail in “Material and Methods” would be very useful (sampling design, photos vs collection of corals, surface utilized for LCC, kind of sampling, etc. ...). If the paper uses only literature information, then it should be labelled as “review” rather than “article”.

Authors Response: Thank you for the suggestion. The data used in the ms are already published and some not yet. We have included information about origin of data, methods and way the figures were made in the figure legends and cite the original reference. We have submitted this manuscript as a “Review” and not as an “Article” and this will be clarified with the journal

The ms is acceptable for publication, provided that a number of minor changes are introduced. Surely, the ms would benefit from much attention to editing and readability of figures and tables.

Minor remarks:
In the affiliation addresses list, the telephone number of the corresponding author must be completed or

deleted. Is the address e-mail@e-mail.com for all different institutions right?

Authors Response: Those email after institutions were a mistake, it is now removed

Line 26: change “whrein” in “where”.

Authors Response: corrected

Line 36: put “Durusdinium” in italics.

Authors Response: changed to italics 

Line 68: at least another paper dealing with bleaching in the Indian Ocean should be cited. I suggest Montefalcone et al., 2018. Glob. Change Biol. 24: 5629-5641, doi.org/10.1111/gcb.14439 (the references therein may also be of interest).

Authors Response: The paper mentioned by reviewer doses not deal with bleaching in the Indian Ocean per se. They also mention references for the bleaching that occurred in 2015 and 2016 and the reference that we have used does the same. But, we agree with the reviewer and included this reference in the revised version 

Line 96: reword to “Resilience, a theory introduced to describe how ecosystems respond to disturbances [29],”

Authors Response: The sentence has been corrected

Lines 96-98: discuss and add some more recent references on the topics of resilience, disturbance and phase shift.

Authors Response: This part of the manuscirpt has been revised

Lines 226-227: add some detail on diversity in different locations and explain how you measured H’. What about species distinction, their number/cover, sampling area, etc. ...

Authors Response: This part has been revised. The H’ refers to Shannon Wiener’s index, measured along the 10m line intercept transect. The sentence has changed to “There is no significant correlation between species diversity (Shannon-Wiener’s index, H’) per 10m line intercept transect and depth, except at the northern tip of Nanwan, where the species diversity decreases with depth”

Line 258: change “[[82]” in “[82]”. Line 370: change “may” in “many”.

Authors Response: reference 82 and table 1 are in same line hence we did not change anything. “may has been changed to “many” 

Figures and Tables: take care of the editing of the figures and ensure that the characters are legible once published. In particular:

Figure 1: in the map, add Current” after “Kuroshio”.

Authors Response: We added “current” in the revised figure.

Figure 2: the map and the photos are too small; improve the readability of the

abbreviations and words in the map.

Authors Response: Sorry, the original figure reads well in all aspects. The problem is due to reduced resolution in the manuscript for review as well as small size. We are going to provide original high resolution figure files and also request to make the figure size big.

Figure 3: improve the readability.

Authors Response: same as above (for figure 2 comments)

Figure 5: make bigger.

Authors Response: Ok, we will suggest the journal to keep all the figures in big size, so that they are clear

Figure 6: make bigger the legend.

Authors Response: The Figure has been revised

Figure 7: make the map bigger and the characters on the axes more legible; make the graph bigger.

Authors Response: Ok, we will suggest the journal to keep all the figures in big size, so that they are clear

Figure 8: in the legend change the name “Cloadopoium” in “Cladocopium”. There is the place to enlarge a bit the figure.

Authors Response: Ok, we will suggest the journal to keep all the figures in big size, so that they are clear

Table 1: the characters are quite unreadable.

Authors Response: Table 1 has been revised

Table 2: use a bigger character. Change “limitted” in “limited”. I suggest merging the columns Category and Name, such as: Typhoon Peggy, Temperature anomaly bleaching, Amorgos ship grounding, Sea anemone Condylactis sp outbreak, etc.; by the way, why you do explicit that Condylactis is a sea anemone but do not reveal that Codium edule is a green alga? Uniform the editing of the column “year”.

Authors Response: Table 2 has been revised

Table 4: specify the unit of temperature data.

Authors Response: Table 4 has been revised

References:

The reference “Kohler & Gill 2066” in Table 1 is not reported in the references list. Add this reference in the list and renumber accordingly the references in the list.

In the reference list, the following reference is mentioned twice. Delete the reference 54 and renumber accordingly the references in the list.

Hughes, T.P.; Graham, N.A.J.; Jackson, J.B.C.; Mumby, P.J.; Steneck, R.S. Rising to the challenge of 543 sustaining coral reef resilience. Trends Ecol. Evolut. 2010. 25, 633–642. 544

54 Hughes, T.P.; Graham, N. A. J.; Jackson, J. B. C.; Mumby, P. J.; Steneck, R. S. Rising to the challenge of 591 sustaining coral reef resilience. TREE. 2010. 25, 633-642. 592

In the reference list, the following reference seems mentioned twice: authors, title, volume and pages are the same, only the year of publication is different. The year in the reference 61 is wrong: delete this reference and renumber accordingly the references in the list.

Lee, H-J.; Chao, S-Y.; Fan, K-L.; Kuo. T-Y. Tide-Induced Eddies and Upwelling in a Semi-enclosed Basin: Nan Wan. Estuar. Coast. Shelf Sci. 1997. 49, 775–787. Lee, H-J.; Chao, S-Y.; Fan, K-L.; Kuo, T-Y. Tide-Induced Eddies and Upwelling in a Semi-enclosed Basin: Nan Wan. Estuarine, Coast. Shelf Sci. 1999b. 49, 775–787.

Authors Response: Excuse us for this mistake, all the references have been re-checked and corrected.

Reviewer 3 Report

This manuscript represents an impressive effort to synthesize long term monitoring of coral reef systems in the Kenting National Park of Taiwan with a focus on resilience under multiple stressors.  There is much interesting data on hydrographic conditions and changes in coral cover, species composition and diversity of symbiotic zooxanthellae and relative abundance under the impact of a nuclear power plant.  The descriptions of the diversity of symbionts relative to coral species and site are particularly interesting and certainly an area that suggests further study and monitoring. One of the points the authors spend time on is the potential role of localized upwelling from storms and tidal currents that may moderate the negative impact of water temperatures due to climate change.  However, there is a negative trend in percent coral cover at 5 to 10 m over the time period of this monitoring (manuscript lines 252 and 253), which deserved a more detailed explanation.  If upwelling of cooler temperatures is going to be contributing to resilience, then the trend in coral cover referenced is troubling.  Related to declining coral cover are changes in relative abundance of different coral genera and Figure 6 is too detailed and small to be of value without greater discussion.

There are a series of misspelled words and a few sentences that are in need of editing (wherein in line 26 and also line 36 of the abstract).  The authors could have devoted a few paragraphs to describing the multiple stressors rather than just listing them. 

In summary, this is a very ambitious undertaking and it could be a most valuable contribution with some refocusing on a few aspects, i.e. Stressors, changes in coral composition, symbiont diversity and editing for spelling.  I hope the manuscript can be tightened up a bit and published as there is much of value in the contents.

Author Response

Response to Reviewer 3 Comments

Reviewer 3

Comments and Suggestions for Authors

This manuscript represents an impressive effort to synthesize long term monitoring of coral reef systems in the Kenting National Park of Taiwan with a focus on resilience under multiple stressors.  There is much interesting data on hydrographic conditions and changes in coral cover, species composition and diversity of symbiotic zooxanthellae and relative abundance under the impact of a nuclear power plant.  The descriptions of the diversity of symbionts relative to coral species and site are particularly interesting and certainly an area that suggests further study and monitoring.

Authors Response: Thank you for your comments

One of the points the authors spend time on is the potential role of localized upwelling from storms and tidal currents that may moderate the negative impact of water temperatures due to climate change.  However, there is a negative trend in percent coral cover at 5 to 10 m over the time period of this monitoring (manuscript lines 252 and 253), which deserved a more detailed explanation.  If upwelling of cooler temperatures is going to be contributing to resilience, then the trend in coral cover referenced is troubling.  Related to declining coral cover are changes in relative abundance of different coral genera and Figure 6 is too detailed and small to be of value without greater discussion.

Authors Response: Thank you for your comments. Upwelling contributes to resilience by cooling water temperature which prevents the occurrence of KNP wide severely bleaching events. The negative trend in coral cover was mainly contributed by repeated typhoons, although it also contributes to resilience by cooling the warm sea surface temperature. An extra paragraph has been added to explain this complicate system in detail.

The legend of Figure 6 has been modified as suggested.

There are a series of misspelled words and a few sentences that are in need of editing (wherein in line 26 and also line 36 of the abstract).  The authors could have devoted a few paragraphs to describing the multiple stressors rather than just listing them.

Authors Response: The spelling mistakes have been corrected and re-checked. We did thing of describing the multiple stressors in detail, but reference no. 1 cited has described multiple stressors in detail. Rather than devoting additional paragraphs on this, which will make already long manuscript more longer, we direct readers to the references mentioned for further reading 

In summary, this is a very ambitious undertaking and it could be a most valuable contribution with some refocusing on a few aspects, i.e. Stressors, changes in coral composition, symbiont diversity and editing for spelling.  I hope the manuscript can be tightened up a bit and published as there is much of value in the contents.

Authors Response: Thank you for the comments